# Importance of Urodynamic Dysfunctions as Risk Factors for Recurrent Urinary Tract Infections in Patients with Multiple Sclerosis

**Miguel Vírseda-Chamorro [1],\***, **Jesús Salinas-Casado [2]** and **Jorge Matias-Guiu [3]**

1 Servicio de Urología, Hospital Nacional de Parapléjicos, Finca la Peraleda s/n, 45071 Toledo, Spain
2 Department of Urology, Hospital Clínico San Carlos, 28040 Madrid, Spain
3 Department of Neurology, Hospital Clínico San Carlos, 28040 Madrid, Spain
* Correspondence: mvirsedachamorro@yahoo.com; Tel.: +34-9254-7751

**Abstract:** Objective: To analyze the role of urodynamic dysfunctions as risk factors for recurrent urinary tract infections (rUTIs) in patients with multiple sclerosis (MS). Material and methods: We conducted a prospective cohort study of 170 patients with MS who underwent a urodynamic study due to lower urinary tract symptoms. Patients were followed for one year, and 114 (84 women [74%] and 30 men [26%]; mean age 49 years) completed the study. Clinical variables and urodynamic findings (free uroflowmetry, cystometry, and pressure-flow study results) were recorded. Results indicated rUTIs was present in 37 patients (32%). Statistical analysis was performed using Fisher's exact test, chi-square test, Student's t-test, and multivariate regression analysis. Results: In univariate analysis, significant differences were observed between patients with and without rUTIs for the following clinical variables: symptom progression time, MS duration, Expanded Disability Status Scale score, and MS type. Regarding urodynamic findings, significant differences were observed in maximum flow rate (Qmax) (lower in patients with rUTIs), voided volume, bladder voiding efficiency, stress urinary incontinence (SUI) (greater rUTI frequency in affected patients), detrusor pressure at maximum flow, and bladder contractility index score. Multivariate analysis identified the urodynamic factors: low Qmax [Odds Ratio (OR) = 0.90 and SUI (OR = 2.95) as the independent predictors of rUTs. Conclusions: Two urodynamic variables: Qmax and SUI, are independent risk factors for rUTIs in MS patients. These two variables might be associated with Pelvic floor dysfunctions.

**Keywords:** multiple sclerosis; recurrent urinary tract infections; urodynamics; urinary bladder neurogenic; pelvic floor





## 1. Introduction

Multiple sclerosis (MS) is the most frequent cause of non-traumatic spinal cord injury [1]. MS has a high prevalence of neurogenic lower urinary tract dysfunction (NLUTD) due to lesions affecting urinary tract innervation [2]. Guidelines on NLUTD [3] consider mandatory urodynamic studies in these patients to document lower urinary tract dysfunctions ("LUTD"). However, despite the great prevalence of LUTD, guidelines on MS do not recommend urodynamic studies unless there are common urologic complications [1,2].

Most MS patients suffer from lower urinary tract symptoms and sexual dysfunction at some stage of the disease. Many efforts have been made to study and treat urogenital problems in MS. These complaints should be brought to the attention of treating neurologists, particularly since many of these problems are currently susceptible to improvement with symptomatic treatment. Adequate treatment can prevent further complications, resulting in an overall increase in the quality of life for MS patients.

It has been reported that there is a correlation between the duration of MS and disability. In the early stage of the disease, a low percentage of MS patients have urological complaints, sometimes associated with other neurological symptoms, although there may

be evidence of urological dysfunction in clinically silent patients. During the course of the disease, most patients with MS develop urinary symptoms, women and men being equally affected. Symptomatic voiding dysfunction was present in most of the patients, such as urgency, frequency, incontinence, and hesitancy-retention. The best-known series addressing urinary symptoms in multiple sclerosis. The differences among series are at least in part due to methodological factors, e.g., cohorts were from different hospital settings and different stages of MS.

Five possible urodynamic patterns can be found in MS [3]:

(a)  Detrusor overactivity without obstruction
(b)  Detrusor overactivity with outlet obstruction (detrusor external sphincter dyssynergia—DSD)
(c)  Detrusor overactivity with impaired contractility
(d)  Detrusor areflexia
(e)  Normal function.

The most common urodynamic finding is detrusor overactivity, the bladder contracts involuntarily, and the person experiences a feeling of imminent micturition. Incontinence can be avoided sometimes by contracting the pelvic muscles, but the pressure can be so great that, despite the contraction of the external sphincter, urine escapes from the bladder, resulting in urge incontinence. Urodynamics might not be necessary for a patient with urinary urgency and mild to moderate paraparesis because symptoms in the bladder of an MS patient can be reasonably assumed to be due to detrusor overactivity that responds to anticholinergics.

Bladder overactivity and DSD often coexist. In this case, both the external sphincter and the detrusor contract involuntarily at the same time. There is hesitancy, an interrupted urinary stream, and incomplete bladder emptying. DSD is very frequent in MS, affecting approximately two-thirds of patients [3]. Detrusor areflexia is uncommon in MS [4].

The goals of MS are to increase the interval between micturition, complete bladder emptying, reduce incontinence, prevent urinary tract infections as well as urological complications related to detrusor external sphincter dyssynergia, high detrusor filling pressures (>40 cm $H_2O$) and presence of an indwelling catheter. All these factors predispose upper tract problems (10% of patients), including vesicourethral reflux, bladder, and kidney stones, hydronephrosis, pyelonephritis, and renal insufficiency.

In every MS patient, a detailed clinical history should be obtained, with emphasis on symptoms of urgency, frequency, incontinence, hesitancy, retention, and nocturia. A voiding diary, including frequency and 24-h urine volume, should be recorded. Postvoid residual (PVR) should be recorded, preferably by ultrasonography, but if this is not available, it can be performed by catheterization after spontaneous voiding, measuring the residual urine.

Laboratory tests should include urine analysis, culture and sensitivity, postvoid residual, urodynamics, voiding cystogram, and occasionally a cystoscopy. The non-invasive tests might be performed annually or more frequently if needed. Invasive tests should be performed under formal expert urological supervision but are only occasionally needed. Indications for urodynamic evaluation are monitoring of voiding pressures, evaluation of symptoms (frequency, urgency, incontinence), large postvoiding residual volumes of urine (retention), recurrent urinary tract infections, deterioration of upper tracts, and evaluation and monitoring of pharmacotherapy.

Urinary tract infections (UTIs) are the main urological complication in MS patients. This complication seriously compromises their health, representing the second most frequent cause of hospitalization in MS patients and doubling their risk of mortality [5]. Furthermore, recurrent urinary tract infections (rUTIs) promote MS relapse by triggering autoimmune processes [6].

In the general population, risk factors for UTIs are thought to include intrinsic clinical factors (e.g., older age, male sex) and extrinsic factors (e.g., prior antibiotic use, urinary catheters, disease severity) [7]. In patients with NLUTD, urodynamic alterations may

promote UTIs [8]. Therefore, expert committees recommend performing urodynamic studies in MS patients with UTIs [9,10]. However, at present, there is not enough empirical data to prove that LUTD may be a risk factor for UTIs in patients with MS.

The objective of this study is to identify the urodynamic signs that may constitute a risk factor for rUTIs in a cohort of patients with MS. Since there are also clinical risk factors, we decided to use a multivariate analysis to determine if urodynamic dysfunctions are independent risk factors.

## 2. Materials and Methods

In this prospective longitudinal cohort study between January 2017 and September 2018, we recruited 170 patients with MS who underwent a urodynamic study due to lower urinary tract symptoms (LUTS). MS was diagnosed by the neurology department in accordance with the McDonald criteria. Inclusion criteria were MS diagnosis time of at least one year and age ≥18 years. We excluded patients with congenital urinary tract alterations, urolithiasis, genitourinary tumors, and NLUTD not secondary to MS. Fourteen patients were excluded due to these criteria. A one-year follow-up was made by the hospital's neurology department, and finally, 114 patients (84 women [74%] and 30 men [26%]; mean age 49 ± 10.0 years) completed the study. The study protocol was approved by our Institutional Review Board, and informed consent was obtained from all patients.

The sample size was calculated based on the data provided by Wiedemann et al. [11] and Bemelmans et al. [12]. Assuming an Expanded Disability Status Scale (EDSS) score difference of 1.5 points, a standard deviation (SD) of 2.36 points, an alpha level of 5%, and a statistical power of 80%, the minimum sample size was calculated at 37 patients per group.

When the follow-up finished, we recorded data regarding demographic, neurological status (including the level of disability, measured using the EDSS [13]), and the presence and type of LUTS and rUTIs diagnosed by the European Association of Urology criteria. Thirty-seven patients (32%) had rUTIs. Therefore, the number of patients in the case group was within the established limits.

The urodynamic study was performed in accordance with the specifications of the International Continence Society (ICS) [14] and guidelines for good urodynamic practice [15], using a Solar polygraph (MMS, Enschede, the Netherlands). The study comprised a free uroflowmetry test, followed by a multichannel urodynamic study including cystometry, a pressure-flow study, and perineal electromyography (EMG). The filling was performed using a 6 Fr double-lumen catheter with saline at ambient temperature and at a rate of 20 mL/min. The voiding phase started when patients referred a strong desire to void or involuntary detrusor contractions occur. Bladder pressure was measured using the urethral catheter. Abdominal pressure was measured using a rectal balloon catheter. Detrusor-sphincter dyssynergia was defined as increased EMG activity during involuntary detrusor contraction. Bladder voiding efficiency (BVE) was calculated as the percentage of post-void residual volume against total bladder capacity. Detrusor contractility was calculated using the bladder contractility index (BCI), and urethral resistance using the bladder outlet obstruction index (BOOI)

Results were stored electronically; the SPSS statistics software (version 20) was used for statistical analysis. All values are expressed as means ± SD. The Fisher exact test and the chi-square test were used for qualitative variables; the t-test was used to compare the means of parametric data. Quantitative data were tested for normal distribution using the Kolmogorov-Smirnov test. Stepwise logistic regression was used to determine the contributing factors for recurrent urinary tract infections. A value of $p < 0.05$ was statistically significant.

## 3. Results

### 3.1. Descriptive Statistics

The Mean MS progression time was 17 ± 8.3 months. Sixty-nine patients (61%) had relapsing-remitting MS (RRMS), 13 (11%) had primary progressive MS (PPMS), 31

(27%) had secondary progressive MS (SPMS), and one (1%) had tumefactive MS. Thirty-three patients (29%) were receiving no treatment. Forty-four patients (39%) were being treated with first-line drugs: glatiramer acetate 8 (7%), interferon beta-1a 7 (6%), interferon beta-1b 7 (6%), peginterferon beta-1a one patient (1%), dimethyl fumarate 5 (4%), and teriflunomide 16 (14%). Thirty-seven patients (32%) were being treated with second-line drugs: natalizumab, one patient (1%), fingolimod, 11 (10%), and alemtuzumab, 7 (6%). The mean EDSS score was 4.7 ± 2.13 points.

The mean LUTS progression time was 10 ± 5.9 months. LUTS comprised filling symptoms in 65 patients (57%), voiding symptoms in 29 (25%), and mixed symptoms in 17 (15%). LUTS type could not be determined in three patients. Five patients (4%) were on clean intermittent catheterization; one patient (1%) used a permanent catheter. The remaining patients did not use any type of urinary catheter.

The free uroflowmetry study revealed a voided volume of 179 ± 185.3 mL, a maximum flow rate (Qmax) of 14 ± 8.7 mL/s, a post-void residual volume of 78 ± 124.8 mL, and a bladder voiding efficiency (BVE) of 25% ± 30.0%.

In filling cystometry, the cystometric capacity was 227 ± 125.5 mL, and bladder compliance was 60 ± 56.3 mL/cm $H_2O$. Detrusor overactivity was observed in 46 patients (40%). The maximum detrusor pressure during involuntary contractions was 39 ± 27.4 cm $H_2O$; the cystometric capacity at first involuntary contraction was 126 ± 89.6 mL. Stress urinary incontinence (SUI) was observed in 23 patients (20%).

In the pressure-flow study, the maximum detrusor pressure (PdetMax) was 41 ± 29.8 cm $H_2O$, the detrusor pressure at maximum flow (PdetQmax) was 27 ± 20.38 cm $H_2O$, BCI was 83 ± 45.3, and BOOI was 5 ± 24.7. Detrusor-sphincter dyssynergia was detected in 37 patients (32%).

### 3.2. Inferential Statistics

Table 1 shows the relationship between clinical variables and rUTI occurrence. Statistically significant differences were observed for symptom progression time (longer in patients with rUTIs), MS duration (longer in patients with rUTIs), EDSS score (higher in patients with rUTIs), and MS type (greater rUTI frequency in PPMS and SPMS).

**Table 1.** Relationship between clinical data and occurrence of recurrent lower urinary tract infections.

| Variable | Patients with rUTIs | Patients without rUTIs | Significance Level |
|---|---|---|---|
| Women/men * | 31 (84%)/7 (16%) | 47 (61%)/30 (39%) | 0.068 |
| Age (years) † | 51 ± 10.8 | 49 ± 9.5 | 0.091 |
| Storage LUTS * | 23 (62%) | 43 (56%) | 0.339 |
| Voiding LUTS * | 10 (27%) | 19 (25%) | 0.494 |
| Mixed LUTS * | 5 (15%) | 12 (16%) | 0.483 |
| Symptom progression time (months) † | 9 ± 6.5 | 6 ± 5.4 | 0.016 ‡ |
| Performing CIC * | 3 (8%) | 2 (3%) | 0.197 |
| MS duration (months) † | 20 ± 9.2 | 16 ± 7.6 | 0.010 ‡ |
| EDSS score † | 5.3 ± 2.11 | 4.4 ± 2.08 | 0.015 ‡ |
| PPMS or SPMS MS type * | 19 (51%) | 25 (32%) | 0.042 ‡ |
| MS treatment * | None: 12 (32%)<br>First line: 13 (36%)<br>Second line 12 (32%) | None: 21 (27%)<br>First line: 31 (41%)<br>Second line:25 (32%) | 0.819 |

* Absolute frequency (percentage). † Mean ± SD. ‡ Statistically significant CIC, clean intermittent catheterization; LUTS, lower urinary tract infection; MS, Multiple sclerosis; PPMS, primary progressive MS; rUTI, recurrent urinary tract infection; SPMS, secondary progressive MS.

Table 2 shows the relationship between urodynamic findings and rUTI occurrence. Significant differences were observed in voided volume (lower in patients with rUTIs), Qmax (lower in patients with rUTIs), BVE (greater post-void residual volume in patients with rUTIs), SUI (greater rUTI frequency in patients with SUI), PdetQmax (lower in patients with rUTIs), and BCI score (lower in patients with rUTI).

**Table 2.** Relationship between urodynamic study findings and occurrence of recurrent lower urinary tract infections.

| Variable | Patients with rUTIs | Patients without rUTIs | Significance Level |
|---|---|---|---|
| Voided volume (mL) † | $103 \pm 135.3$ | $216 \pm 195.5$ | 0.000 ‡ |
| Maximum flow rate (mL/s) † | $9 \pm 7.8$ | $15 \pm 8.6$ | 0.002 ‡ |
| Post-void residual volume (mL) † | $73 \pm 84.7$ | $80 \pm 137.5$ | 0.416 |
| Bladder voiding efficiency (%) † | $35 \pm 37.0$ | $21 \pm 26.2$ | 0.026 ‡ |
| Cystometric capacity (mL) † | $208 \pm 124.8$ | $235 \pm 125.8$ | 0.145 |
| Bladder compliance (mL/cm $H_2O$) † | $62 \pm 64.9$ | $58 \pm 51.8$ | 0.362 |
| Detrusor overactivity * | 12 (26%) | 25 (38%) | 0.135 |
| Maximum detrusor pressure during involuntary contraction (cm $H_2O$) † | $44 \pm 22.9$ | $36 \pm 29.2$ | 0.339 |
| Stress urinary incontinence * | 12 (52%) | 25 (28%) | 0.030 ‡ |
| Maximum detrusor voiding pressure (cm $H_2O$) † | $36 \pm 27.1$ | $44 \pm 30.9$ | 0.093 |
| Detrusor pressure at maximum flow (cm $H_2O$) † | $22 \pm 19.8$ | $29 \pm 20.3$ | 0.046 ‡ |
| BCI † | $72 \pm 45.3$ | $88 \pm 44.6$ | 0.045 ‡ |
| BOOI † | $3 \pm 27.1$ | $6 \pm 23.5$ | 0.278 |
| Detrusor-sphincter dyssynergia * | 14 (32%) | 20 (29%) | 0.222 |

* Absolute frequency (percentage). † Mean $\pm$ SD. ‡ Statistically significant CIC, clean intermittent catheterisation; LUTS, lower urinary tract infection; MS, Multiple sclerosis; PPMS, primary progressive MS; rUTI, recurrent urinary tract infection; SPMS, secondary progressive MS.

To perform a logistic regression analysis, clinical and urodynamic variables with a signification level below 10% in univariate analysis were included in the model (Table 3). The final model identified only two variables that were independently associated with the occurrence of rUTIs. These were the urodynamic variables Qmax [Odds Ratio (OR) = 0.90] and SUI (OR = 2.95) (Table 4).

**Table 3.** Initial logistic regression model.

| Variables | ß | *p* | $R^2$ |
|---|---|---|---|
| MS type | 0.595 | 0.430 | |
| Maximum Flow rate | $-0.110$ | 0.081 | |
| Sex | $-1.224$ | 0.267 | |
| MS duration | 0.053 | 0.427 | |
| Symptoms progression time | 0.041 | 0.631 | |
| EDSS score | $-0.139$ | 0.405 | 0.357 |
| Bladder voiding efficiency | 0.003 | 0.405 | |
| Stress urinary incontinence | 1.495 | 0.037 | |
| Maximum detrusor voiding pressure | 0.016 | 0.454 | |
| Detrusor pressure at maximum flow | | | |
| BCI | $-0.003$ | 0.785 | |
| Constant | $-0.849$ | 0.564 | |

ß, multivariate logistic regression coefficient; *p*, significance level (Wald); $R^2$, determination coefficient; BCI, bladder contractility index.

**Table 4.** Final logistic regression model.

| Variables | ß | *p* | $R^2$ |
|---|---|---|---|
| Maximum Flow rate | $-0.110$ | 0.081 | 0.407 |
| Stress urinary incontinence | 1.495 | 0.037 | |

ß, multivariate logistic regression coefficient; *p*, significance level (Wald); $R^2$, determination coefficient; BCI, bladder contractility index.

## 4. Discussion

Urodynamic risk factors for UTIs in patients with MS have received little attention in the literature, and most studies have not found any urodynamic risk factor. For instance,

Castel-Lacanal et al. [16]. did not observe any relationship between urodynamic findings and urinary complications (including pyelonephritis and urosepsis). However, as not all patients in this study underwent urodynamic testing, its results are not reliable. Naki-poglu et al. [17] neither identify the association between urodynamic data and UTIs in patients with MS. However, these researchers provide very little data; their urodynamic diagnoses include "detrusor hyporeflexia" (which is not defined in the ICS terminology) and do not describe how this diagnosis was made. Only Gallien et al. [18] reported that post-void residual volume was a risk factor for pyelonephritis but not for lower urinary tract infections.

Conversely, our study identified a series of urodynamic risk factors for rUTIs in univariate analysis. These factors were voided volume (lower in patients with rUTIs), Qmax (lower in patients with rUTIs), BVE (greater post-void residual volume in patients with rUTIs), SUI (greater rUTI frequency in patients with SUI), PdetQmax (lower in patients with rUTIs), and BCI (lower in patients with rUTIs).

Arya et al. [19] also reported lower voided volume in women with rUTIs. That article attributes this to reduced bladder capacity due to a lower threshold of bladder sensitivity in patients with rUTIs. In our study, patients with rUTIs also had lower cystometric capacity, although this reduction was not statistically significant. These differences may be less pronounced in our series because it included both men and women.

Post-void residual urine is a risk factor for UTI in patients with NLUTD because it favors the rapid accumulation of bacteria that would otherwise be removed during voiding [20]. Although we found no association between rUTIs and post-void residual volume in absolute terms, post-void residual volume as a percentage of total bladder capacity (BVE) was significantly greater in patients with rUTIs. Detrusor underactivity results in a low PdetQmax and post-void residual urine. In our study, patients with rUTIs had lower BCI and PdetQmax compared with patients without infections. Van Batavia et al. [21] also concluded that children with detrusor underactivity were more likely to have a history of infection. This relationship was attributed to urinary stasis, which predisposes to UTIs.

However, the main finding of our study was that multivariate analysis showed that clinical risk factors and other urodynamic factors depend on two independent urodynamic risk factors: Qmax and the presence of SUI. In other words, clinical risk factors for rUTIs are the consequence of urodynamic dysfunctions.

Qmax was significantly lower in patients with rUTIs; Athanasiou et al. [22] also observed a lower maximum flow rate in women with rUTIs. A low Qmax may result from dysfunctional voiding. Dysfunctional voiding disrupts the laminar urinary flow through the urethra, causing UTIs as bacteria are transferred back from the meatus to the bladder because of the milk-back phenomenon [23]. Our study did not find any relationship between detrusor-sphincter dyssynergia and rUTIs. Nevertheless, perineal EMG is not accurate enough to measure external sphincter activity during voiding [24]. Studies have revealed that MS patients have poor control of their pelvic floor, which can lead to dysfunctional voiding. Pelvic floor muscle training significantly improves Qmax in people with MS [25].

The significantly higher frequency of rUTIs in patients with SUI is also related to pelvic floor dysfunction. Moreover, Ahmed et al. [26] found in women of childbearing age that low urinary flow and genital prolapse are independent risk factors for rUTs. In this study, urinary incontinence and genital prolapse were observed more frequently in patients with rUTIs. Altered anatomy may predispose to rUTIs in women due to a shorter distance between the anus, vagina, and urethra. The power of the pelvic floor is reduced in patients with MS. Pelvic floor muscle training improves SUI in these patients with a significant reduction in 24-hr pad testing [27]. Therefore, there is a possible common point in both urodynamic dysfunctions: an alteration of the pelvic floor. Should other studies confirm the relationship between UTIs and pelvic floor dysfunction, its rehabilitation may be a way to prevent urinary tract infections in MS patients.

Treatment must be individualized for every patient [4]. Treatment has to be adapted to the phase of the disease, the symptoms, and the results of non-invasive tests. Possible causes of clinical exacerbation, such as urinary tract infections, should be prevented and treated as soon as possible. When urodynamic studies are not considered necessary or not immediately available le, patients should be treated conservatively, using timed voiding, limiting fluid intake, suprapubic tapping to trigger micturition, clean intermittent catheterization, and use of diapers. In a disease as variable over time as MS, irreversible or ablative surgery should be avoided unless it has been confirmed that the voiding pattern and urodynamic findings have stabilized over time. In case of doubt, specific treatment strategies should be based on urodynamic findings because starting or changing medication without urodynamics can sometimes induce more harm than benefit. A permanent indwelling catheter should be avoided whenever possible.

The gynecologist Arnold Kegel postulated pelvic floor exercise in the 1950s for the treatment of postpartum stress urinary incontinence [28]. Subsequently, these exercises have been proposed for the treatment of other urinary disorders, such as detrusor overactivity or overactive bladder syndrome. In this case, the reinforcement of the tone of these muscles would produce an inhibitory effect on the parasympathetic responsible for the involuntary contractions of the detrusor. In this case, the reinforcement of the tone of these muscles would produce an inhibitory effect on the parasympathetic responsible for the involuntary contractions of the detrusor. However, perineal physiotherapy is not only based on increasing the tone of this muscle. However to modulate or reduce it in the case of voiding dysfunction syndrome such as the so-called Hinman syndrome. Its use in patients with neurogenic detrusor dysfunction is more controversial [29].

Some authors believe that, due to the lack of pudendal control in these patients, this treatment is not effective. However, other authors consider that providing the innervation lesion is not complete, its use in these patients is advisable. We have no evidence of its usefulness in patients with multiple sclerosis. New studies are needed to verify its usefulness in this type of patient [30].

The main limitation of this study is that our results might not be extrapolated to the general population since patients referred to urodynamic evaluation are more likely to have lower urinary tract symptoms than patients of the general population. However, since there is a sizable percentage of patients with MS with lower urinary tract symptoms2, the external validity of our study is preserved. Conversely, the main strength of the study is its prospective characteristic, the fact that all patients underwent a comprehensive urodynamic study and the multivariate analysis, enabling us to establish for the first time that the independent risk factors for rUTIs in patients with MS are urodynamic and probably related to pelvic floor dysfunction.

## 5. Conclusions

Urodynamic pathologic signs of voiding dysfunction are risk factors for rUTIs. Clinical risk factors and other urodynamic factors are dependent on two independent urodynamic risk factors: diminished urinary flow and the existence of SUI. These findings suggest that pelvic floor dysfunction might play a role in MS patients with rUTIs.

**Author Contributions:** Conceptualization, J.S.-C. and J.M.-G.; methodology, M.V.-C.; software, M.V.-C.; validation J.S.-C. and J.M.-G.; formal analysis, J.S.-C.; investigation, M.V.-C.; resources, J.M.-G.; data curation, J.M.-G.; writing—original draft preparation M.V.-C.; writing—review and editing, J.M.-G.; visualization, J.S.-C.; supervision, J.S.-C. All authors have read and agreed to the published version of the manuscript.

**Funding:** This research received no external funding.

**Institutional Review Board Statement:** The study was conducted according to the guidelines of the Declaration of Helsinki and approved by the Ethics Committee of Hospital Clínico de Madrid.

**Informed Consent Statement:** Informed consent was obtained from all subjects involved in the study.

**Data Availability Statement:** Full data will be available upon reasonable request to the corresponding author.

**Conflicts of Interest:** The authors declare no conflict of interest.

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
