# Peer review of "Importance of Urodynamic Dysfunctions as Risk Factors for Recurrent Urinary Tract Infections in Patients with Multiple Sclerosis"

_2673-4397, doi:10.3390/uro3010011_

Round 1

Reviewer 1 Report

I read with great interest this paper. I have only minor comments:

- please define the type of the study. I suppose a longitudinal cohort study. However, please improve this section.

- please make it easier to understand the sample size calculation. In the first phase, you included more than 170 patients but the sample size identify 31 patients per group. Please improve this aspect.

Author Response

Y

Reviewer 2 Report

Thank you for giving me the opportunity to review this manuscript. 

The Authors propose a perspective cohort study to correlate urodynamic signs and UTIs in patients with multiple sclerosis. 

The work is interesting, its contents are effective and the methodology is correct. Overall, the manuscript is scientifically sound. 

Therefore, I would offer them some formal suggestions just to improve the readability and some little improvements in english form are required (Line 46, line 55, line 253, ecc). 

ABSTRACT

Line 16: "results indicated rUTI". I would suggest to change with "rUTI were present in 37 patients", just to avoid confusions with the use of the word "results".

Line 26: the phrase needs to be rewritten in a more appropriate form. "Qmax" should be replaced with "low Qmax". Then, avoid the repeat of "variables" in two consecutive phrases.

INTRODUCTION

Line 39: "higher" is "high".

Line 42:  after LUTD place ","

Line 43: "there are urologic complications" may be enlighten with "they are common".

Line 56: "Our hypothesis [...] the case" this phrase may be completerly deleted, since it makes heavy the reading flow. 

Line 59: "urodynamic risk factor" is an inappropriate expression. I think the Authors would say "urodynamic signs that may constitute a risk factor". 

Line 60: "because"-> since

MATERIALS AND METHODS

Line 67: what do you mean with "MS progression time of at least one year"?

Line 75-79: I find difficult to understand this paragraph, please add more information or contextualize it. 

Line 81 "MS was diagnosed by the neurology [...] McDonald criteria": I suggest to move it up to line 67, after "(LUTS)".

- Question 1: why the Authors did not collect data about patients drug assumption? They referred only about MS therapy

RESULTS

- Table 1: why men do not appear?

CONCLUSIONS

Line 268: "Urodynamic dysfunction" may be replaced with a more correct "urodynamic pathologic reliefs (or signs) of voiding disfunction". The Authors may improve the terminology on this side. 

Overall, the Authors may improve this section making easier the articulation of lines 268-270.

FINAL CONSIDERATIONS

The work merits to be considered for its publication on Uro, nevertheless it requires an improvement of its presentation and a minor english spell check.

I remain at your disposal for any concern.

Best Regards

Author Response

Y
